# Organ-On-A-Chip Database Revealed—Achieving the Human Avatar in Silicon

**DOI:** 10.3390/bioengineering9110685

**Published:** 2022-11-12

**Authors:** Lincao Jiang, Qiwei Li, Weicheng Liang, Xuan Du, Yi Yang, Zilin Zhang, Lili Xu, Jing Zhang, Jian Li, Zaozao Chen, Zhongze Gu

**Affiliations:** 1State Key Laboratory of Bioelectronics, School of Biological Science and Medical Engineering, Southeast University, SiPaiLou # 2, Nanjing 210096, China; 2School of Life Science and Technology, Southeast University, Nanjing 210096, China; 3Jiangsu Avartarget Biotechnology Corp., Suzhou 215163, China

**Keywords:** organs-on-a-chip, organoid, database, mathematical modeling, personalized medicine

## Abstract

Organ-on-a-chip (OOC) provides microphysiological conditions on a microfluidic chip, which makes up for the shortcomings of traditional in vitro cellular culture models and animal models. It has broad application prospects in drug development and screening, toxicological mechanism research, and precision medicine. A large amount of data could be generated through its applications, including image data, measurement data from sensors, ~omics data, etc. A database with proper architecture is required to help scholars in this field design experiments, organize inputted data, perform analysis, and promote the future development of novel OOC systems. In this review, we overview existing OOC databases that have been developed, including the BioSystics Analytics Platform (BAP) developed by the University of Pittsburgh, which supports study design as well as data uploading, storage, visualization, analysis, etc., and the organ-on-a-chip database (Ocdb) developed by Southeast University, which has collected a large amount of literature and patents as well as relevant toxicological and pharmaceutical data and provides other major functions. We used examples to overview how the BAP database has contributed to the development and applications of OOC technology in the United States for the MPS consortium and how the Ocdb has supported researchers in the Chinese Organoid and Organs-On-A-Chip society. Lastly, the characteristics, advantages, and limitations of these two databases were discussed.

## 1. Introduction

Organs-on-a-chip (OOCs), which are also called microphysiological systems (MPSs), are an emerging technology [1]. OOCs are physiological organ bionic systems based on microfluidic chips. Microfluidic chips have accurately etched micron channels, controlled microfluidics, and provide a more suitable and simulated context by regulating key parameters for cell and tissue culture [2,3,4,5,6]. Drug development and screening require a rigorous and long testing process [7]. Traditionally preclinical trials have mainly been used in two-dimensional in vitro cell models and animal models [8,9]. However, two-dimensional in vitro cultures have limitations in terms of the specific differentiation of cells and tissues and cannot accurately simulate the in vivo environment [10]. Then, a 3D organoid culture supported by a hydrogel or extracellular matrix (ECM) was developed. However, organoids have limitations regarding their constancy and analysis capabilities [11,12,13]. In vivo models, such as small or large animal models, also experience problems; animals often fail to accurately replicate the human body due to species differences, the cost of animal experiments is high, and they pose ethical problems [14]. As a future alternative to these models, OOCs overcome many of these shortcomings in the fields of drug screening, disease modeling, and toxicity testing (Figure 1). As such, we need to improve the efficiency of drug development and clinical correlation, and to study toxicology and disease mechanisms [2,4,5].

In typical biomedical applications, OOCs usually generate a large amount of data, i.e., image data, readout from sensors, and ~omics data, in the process of research and development (R&D) in both medical and/or pharmaceutical use [4]. Collecting, processing, organizing, and explaining these data is valuable for current researchers and future supervisors from regulatory agencies. Owing to the complexity of these data and information, they cannot be managed only by file and directory systems as this would be inefficient and would not meet the integrity requirements [15]. When the generated data create “information overload” and the existing methods limit the exploration of the data, a computerized database management system (DBMS) should be developed [15]. Of the major types of database management systems, a relational database is a classic data-management choice in the biomedical field [16]. To date, two databases are specific to OOCs. The USA National Institutes of Health (NIH) and the University of Pittsburgh have developed the BioSystics Analytics Platform (BAP) for OOC experimental design, data loading, storage, visualization, etc. The Southeast University and the Chinese Organoid and Organs-on-a-Chip society have developed another organ-on-a-chip database (Ocdb), including literature, patents, related toxics, drugs, and other data [17]. Starting from the classification and structure of OOC, in this study, we discussed the design and development of BAP and Ocdb, as well as their pros and cons.

## 2. Classification and Structure of Organs-On-Chips

A wide variety of OOCs have been designed, and their organs, detection targets, morphology and design may vary. The FDA outlined the design goals of OOCs as accurately recreating the natural physiology and mechanical forces experienced by cells in the human body [18]. Any microfluidic system that meets this design can be considered an OOC and should be included in the OOC database. To facilitate comparison in studies, static two-dimensional culture models are also included in the BAP. The problem is that it clutters the model data and hence the difficulty of retrieval. The solution to this problem is to classify OOCs from different perspectives and label them for retrieval.

Firstly, OOCs can be classified according to their simulated organs, for instance, Liver-on-Chips [19,20], Brain-on-Chips [21,22,23], Cardiac-on-Chips [24,25], Bone-on-Chips [26,27], Kidney-on-Chips [28], Lung-on-Chips [2,29], Muscle-on-Chips [30], Intestine-on-Chips [31,32], Skin-on-Chips [33,34,35], and Multiple-Organs-on-Chips [36,37], etc. This is also the most direct and easiest method of classifying OOCs. Figure 2 shows some of the OOCs that have been developed for various organs of the human body.

### 2.1. Simulation Mode of Organs-On-Chips for Organ Functions

From a macroscopic viewpoint, different organs have different morphologies and functions. However, in current OOC systems, due to technical limitations, the components of different OOC systems are often limited; in other words, the methodology for OOCs construction and simulation usually share common modes and units. Here, we classified the design and construction of current OOCs into four categories to facilitate analysis:

Membranous mode chips could be characterized as the first mode. Their scope covers microphysiological systems, including lung alveoli modeling, white adipose tissue system modeling, kidney proximal tubule modeling, blood–brain barrier modeling, etc. Membranous mode chips originated from the trans-well model that has been commonly used in biological research [39]. It connects two independent chambers through a layer of microporous membrane or semi-permeable membrane for researchers to study the material exchange between the chambers. The blood–brain barrier (BBB) is a typical application of membrane mode OOC [21]. The BBB is a key structure between the central nervous system and other parts of the body. The function of the blood–brain barrier is to promote the entry of required nutrients into the brain and eliminate potentially harmful compounds [40]. Brown J.A. et al. designed a neurovascular unit (NVU) OOC (Figure 3A). The device included a vascular cavity composed of endothelial cells and a brain cavity composed of astrocytes, pericytes, and nerve cells separated by a microporous membrane. The model was verified by fluorescein isothiocyanate (FITC)–glucan diffusion and transendothelial resistance (TEER) experiments [21]. The white adipose tissue (WAT) chip creates a micro-physiological environment supporting the long-term culture of functional adipose tissue. In order to restore the tissue in the body, the system has an endothelial barrier to protect it from blood flow shear stress and convective flow, separate the cell chamber from the medium microchannel, and diffuse the medium through the microporous membrane [41]. Griffith and coworkers’ multiple-organ series chips are also designed based on an improved membrane chip [42]. However, the problems of membrane chips include the difficulty in finding biocompatible supporting materials in the chip, the difficulty in processing and fabricating these porous membranes, and their disparity with real biofilm in the human body.

Multicellular co-culture chips with one or multiple chambers for multicellular co-culture could be characterized as the second mode. The chips allow the cells’ spontaneous self-organization and interaction. This process is also involved in distributing different cells in different connected chambers. Its scope covers liver-on-a-chip, the blood capillary system, bone-on-chip, etc. The Liver Acinus Microphysiology System (LAMPS) is an OOC model of the hepatic acinus [19]. It mixes primary human hepatocytes, human endothelial cells, Kupffer cells, and stellate cells and simulates oxygen concentration zoning in the liver by controlling the flow rate. On the basis of LAMPS, a breast cancer cell polyculture model was established. Based on the metastatic microenvironment of the human liver microphysiology system, estrogen receptor mutation ER+MCF7 cells were added to evaluate its growth [43]. The Mimetas company makes use of the hydrophobic stripes generated by dry etching in the chip to separate several chambers [44,45]. A two-channel, microfluidic OOC culture device was adapted to co-culture human CD34+ cells and bone marrow-derived stromal cells (BMSCs). The two channels of the device are arranged on the top and bottom, connected by a porous membrane in the middle. The co-culture of human CD34+ cells and BMSCs is filled in the top channel while the bottom channel is simulated as vascular perfusion (Figure 3B). Bone marrow on a chip studies the differentiation and maturation of multiple blood-cell lineages, drug toxicity, and symptoms of hematopoietic defects in the rare genetic disorder Shwachman–Diamond syndrome [46]. By adding reactants of gelling in each chamber, the organ-on-chips form 2–3 connected chambers. Therefore, the above chips can be classified into the same category.

Muscle bundle chips is the third mode, includes skeletal muscle-on-a-chip and myocardium-on-a-chip, etc. Madden et al. developed a skeletal muscle culture system (Figure 3C). Human myogenic cells were formed after amplification, hydrogel formation, and low serum medium induction [30]. A platform termed Biowire II enables the generation of chamber-specific cardiac tissues [47]. Electrophysiologically distinct atrial and ventricular tissues can be created on this platform. Two parallel poly (octamethylene maleate (anhydride) citrate) (POMaC) wires allow continuous and simultaneous quantification of force and Ca^2+^ transients. These bundle-based chips mainly measure strong contraction and calcium transient under electrical and pharmacological stimulation.

Mixed-form chips involve multiple components. The majority of organoids-on-chips could be classified into this category, including tumor-on-chips, brain organoid-on-chips, lung-on-chips, etc. Wang et al. developed a brain organoid system on a chip [22]. Compared with brain organoids differentiated in a 3D suspension medium, this OOC model provides the ability to accurately control microenvironment factors. The effects of nicotine exposure on neuronal differentiation and marker expression during the development of brain organoids were discussed by TUNEL detection, cryo-sectioning and immunohistochemistry analysis, real-time polymerase chain reaction (PCR) analysis, neurite outgrowth analysis, etc. (Figure 3D). An increasing number of works have also combined organoids and microfabrication, including 3D bioprinting, to develop a form of OOC containing both the bottom-up methodology (cell self-assembly) and the top-down methodology (micromanipulation process) [48].

Table 1 summarizes the features of the four categories and their representatives. Similar to the development of multi-organ chips, OOCs can simulate multiple patterns, morphologies, or functions. Therefore, some chips may fit into different categories at the same time. For example, the LAMPs not only co-cultures multiple cells in chambers but it also has a porous membrane between chambers. Despite this problem, performing the classification as multiple-choice labels for OOC models informs the way models are retrieved and designed. When setting up the classification of OOC designs, each developer should consider that the structure and physiological functions of the human body are the basis of each type of OOC design, classification, and analysis. As an in vitro organ model, OOCs should simulate (or to some extent reflect) the specific pattern, morphology, and function of a specific human organ.

### 2.2. Device Design and Essential Parts of OOCs

The classification of organ-on-chips according to organs or simulation methods is facilitated through horizontal analysis. There are some basic structural and design principles of microphysiological systems. The standards for OOCs are still being developed. Modularizing the structure of OOCs is beneficial for the standardized production and medical application of OOCs. Additionally, dissecting OOCs helps to better understand and store OOC models (Figure 4).

#### 2.2.1. Manufacturing Process of Device

Microfluidic devices enable OOCs to culture living cells in continuously perfused micron-sized chambers for simulating the physiological functions of tissues and organs [4]. In order to realize cultivation close to the physiological environment in vivo, the chip needs a complex structure to produce physical force, the cyclic supplement of biochemical substances, the interaction between different tissues, and so on. The microfabrication technology of chips in OOC mainly includes lithography and etching [51,52]. In addition, fabrication approaches include hot pressing, injection molding, laser ablation, solid object printing, the LIGA method, etc. [53,54,55]. The chip design requires developers to select the most appropriate processing method according to both the structure itself and the material of the device.

#### 2.2.2. Material of the Device

The material of the chip not only affects the manufacturing process of the device but also is closely related to the performance, convenience, cost, and data detection method of the OOC. There are a great variety of materials used to create microfluidic chips, including monocrystalline silicon, quartz, glass, and organic polymers [56,57,58,59,60]. At present, the most commonly used chip material is polydimethylsiloxane (PDMS). This material has many advantages, including low cost, ease of use, good biocompatibility, and optical clarity to support real-time and high-resolution optical imaging [56]. However, the disadvantages of this material are also obvious. PDMS is a lipophilic material which may absorb many compounds, especially hydrophobic small molecular compounds [61]. Furthermore, the limited capacity for mass production of devices with PDMS also hinders the commercial application of PDMS-based OOCs in the fields of large-scale drug development, personalized medicine, and so on. Alternative materials to replace PDMS focus on organic polymers [56]. The first type is elastomers. PDMS is one type of elastomers, and poly (itaconate-co-citrate-co-octanediol) (PICO) and POMaC polymers used by some cardiac tissue platforms are elastomers [47,62]. The second type is thermoplastic polymers, such as polystyrene (PS) and poly (methyl methacrylate) (PMMA) [47,63]. Hydrogel is another alternative material commonly used with 3D printing technology. Complex topography support for 3D glomerulus fabricating is an application example [64]. In addition to the above materials, there are other inorganic materials that could be used. Silicon has good chemical inertness and thermal stability, but it is fragile and has poor photoelectric properties [57,58]. Quartz and glass have excellent electroosmosis and optical properties, but the price is high [59,60]. Additionally, paper chips have also been in focus due to their simple fabrication, good biocompatibility, and fast analysis speed [65], and they have received increasing attention.

#### 2.2.3. Cell Sources

OOC models need to reproduce certain structures and functions of tissues and organs; thus, appropriate cell sources and cell types should be prepared and provided. Primary cells isolated from specific organs without any gene modification are one of the ideal cell sources for OOCs. These primary cells can demonstrate similar functions as they do in in vivo environments. However, in vitro culture of primary cells could be challenging for certain types, such as neurons and cardiomyocytes, and can only be stably available for a limited time [66]. The development of human-induced pluripotent stem cells (iPSCs) provides a potential alternative for the cell source of OOCs [67,68]. iPSCs can be induced to derive into different organ-specific cells. There are commercially available organ-specific cells derived from iPSCs [69]. iPSCs from patients also carry their genetic information, leading to the feasibility of creating disease models for these patients. Using gene editing technology, mutation-caused specific organ functions can be recreated and studied, which also has broad application prospects [70].

#### 2.2.4. Sensory Systems

OOC platforms require sensor systems to monitor cell activity and function along with the detection of analytes. It is important to integrate sensor systems into OOC devices. However, there are some challenges. Compared with two-dimensional culture, OOC needs to use appropriate approaches to make conventional detection and screening methods achieve the goal of high sensitivity, fast speed, and small volume [69]. Most biochemical reactions on the microfluidic chip platform occur on the micron scale. Furthermore, the real-time and nondestructive signal measurement integrated into the chip can show the dynamic interaction between cells or even between multiple organs [69]. The sensory systems on organ chips could be divided into multiple types: optical detector, electro/chemical detector, mass spectrometry detector, etc. [52,71].

#### 2.2.5. Settings

Microfluidic chips can control environmental parameters that are not easily controlled in traditional 3D static cultures. Control is achieved by device design or bioreactors that add settings in real-time during the cell culture process [4]. For example, the control of perfusion flow rate is one of the most common settings of OOC. Perfusion brings continuous nutrition and shear stress to the model and affects cell cultures and their gene expression profiles [72]. Different flow rates can also control different oxygen concentrations and oxygen tension [19]. In addition, the differentiation and maturation of muscle, nerve tissues, or bone requires mechanical force or electrical stimulation to drive them. Other settings also include pH value, radiation dose, protein concentration, etc.

## 3. Database Design

A database is a collection of specific data that is organized, stored, and managed according to a designed data structure [73]. This chapter describes how to design the data structure of a database with the OOC model as the data subject.

### 3.1. Data Model

A data model defines the structure, organization, and constraints of data. It is a tool and method to extract data, data relationships, and data semantics [73]. When designing the structure of an OOC database, selecting the appropriate data model is the basis. The mainstream data models are logical models, which are divided into hierarchical, mesh, relational, and object-oriented models. Since being proposed in 1970, relational databases have been widely used in various fields. They also provide a classic solution to data storage in the biomedical field [74]. All data and relations in a relational database are represented by tables. The column head of the table is called an attribute. Its value range is the field of the attribute. Each row represents the relationship between a series of values. A table is a collection of relations [74]. Many open-source or commercial relational database management systems are available, such as MySQL, Oracle, SQL Server, etc. New biomedical databases can perform architecture design and deployment based on these mature database management systems [15]. For example, the Colorectal Cancer (CRC) Biomarker Database (CBD) collects and sorts information about CRC biomarkers for researchers and was constructed under MySQL DBMS [75]. The Biologic Specimen and Data Repositories Information Coordinating Center (BioLINCC), which is based on the PostgreSQL database, is a data-sharing center in the field of heart, lung, and blood diseases [76]. Ocdb was developed based on the MySQL database [77]. However, the disadvantage of a relational database is that when the number of data tables increases, the database design paradigm becomes an obstacle to database improvement: improving the database will be difficult, and the query efficiency will become low. Subsequently, a new data storage model named NoSQL was designed. This model has a flexible data structure and does not rely on Structured Query Language (SQL), so it is suitable for large, complex, and dynamic data sets [78,79,80]. At present, some projects have begun to adopt the advanced architecture of the NoSQL database design model. For example, NCBI’s free biomedical literature database PubMed was developed using the open-source system mongo DB and a distributed cloud architecture [81]. BAP was developed with the PostgreSQL database as the core [82]. It is one of the most advanced open-source relational databases and provides support for semi-structured data storage [83]. Compared with the traditional relational database MySQL, PostgreSQL has higher query, storage, and indexing efficiency [79,80].

### 3.2. Entity and Relationship Design

The entity relationship model is an object-based logical model. This model is a conceptual tool to describe the real world with entities, attributes, and relationships [84]. It is usually used to extract things and information of interest in the initial stage of database system design. Then, in the logical design stage, the conceptual model should be implemented into the logical model; that is, the ER model is extended to the relational model. Specifically, entity and association sets are represented by a set of tables [85,86]. Entities are objects with the same characteristics and properties. An attribute is a feature of an entity. An entity can be characterized by a variety of attributes. A relationship is the way in which entities connect to each other [84]. In the concept of the entity relationship model, the world is composed of a series of entities with various attributes and the relationships between these entities [87]. According to the previous OOC classification and structure, an OOC database should contain at least the following entities: OOC, OOC device, device location, cell, detection target, detection method, and settings. OOC and OOC devices, cells, detection targets, detection methods, and settings are many-to-many inclusion relationships, OOC devices and device locations are many-to-many inclusion relationships, and detection targets and detection methods are one-to-many usage relationships. Figure 5 shows the entity relationship diagram. These entities are only for the purpose of storing the information of the OOC models. The OOC database should also store the corresponding experimental information and resulting data of the model or other information helpful to the OOC research.

BAP is designed comprehensively to store data of MPS models. Taking this website as a reference for entity relationship analysis, the above entities can be further subdivided into categories, including MPS model, MPS device, device location, manufacturer, experimental center, cell type, cell source, cell instance, cell biosensor, cell provider, detection category, detection target, detection method, unit, setting, detection provider, and references. In addition, BAP is also a powerful platform for experiment design, data management, and data analysis. A study library is designed in the database, in which OOC studies can be designed, and the assay data can be uploaded and analyzed [88]. BAP also searches and extracts for compounds and clinical data from other existing databases, e.g., PubChem, ChEMBL, DrugBank, ToxNet, and openFDA, and integrates the compound library, which contains chemical properties, biological activity, drug experiments, adverse reports, and other data, to facilitate the database users to design and evaluate drug selection before starting a study [82]. The following figure depicts the E-R diagram developed with reference to the design of BAP.

The MPS model details page of BAP lists the attributes of the model entity, as well as the related location and cell type information. On the model list page, models can be sorted by organ or type, and the search function is also set to quickly find the corresponding models. However, there is a lack of model classification for subdivisions. The corresponding model cannot be found from the perspective of the specific function or mode simulated by the model. The model list page can be visited at https://mps.csb.pitt.edu/microdevices/model/ (accessed on 23 September 2022).

## 4. Developed OOC Database in the World

### 4.1. BAP

#### 4.1.1. Functions of BAP Website

As a web database, BAP adopts a browser/server architecture. The back-end uses the Django framework [89], Anaconda Python [90], and the PostgreSQL database [83], and the front-end uses HTML, CSS3, and JavaScript [82]. The result is an open and accessible internet website with an easy-to-operate icon-driven interface. With the development of the internet, a browser with beautiful and convenient pages is a popular interactive way to lay out open databases. There are different interaction designs for some other databases, such as the cancer database Survey, Epidemiology, and End Results (SEER), which collects the cancer diagnosis, treatment, and survival data of approximately 30% of the U.S. population [91]. Users who want to access data from SEER must download and use the supporting software SEER*STAT or download ASCII binary data packets and process them in other ways to browse the data, which is inconvenient. Medical Information Mart for Intensive Care (MIMIC) provides physiological data such as ECG signals, PPG signals, and arterial blood pressure signals (ABP) collected from Intensive Care Unit (ICU) wards [92]. MIMIC is accessed via the physiological data resource website PhysioNet. Users must select the record sample, signal type, output signal length, and storage type on the website. The downloaded file requires further coding in MATLAB. This is a challenge for users who are mostly clinicians. This graphic-driven interaction design of BAP reduces the user’s threshold and makes it convenient for researchers to access the database.

BAP straightforwardly divides the functions of the website on the main page. The first line is the icon to view the information. There are icons to view the information on the OOC model, compound, and cell, respectively. The second line is a simple workflow: research design–data analysis–computational modeling [82]. For users, the main functions are as follows:(i)Study design.

BAP is not only used to store the information of the developed OOC model but it is also used to manage the experimental data obtained by the model and compare it with the biochemical, preclinical, and clinical data of drugs or to compare the performance of different designs or versions of the same type of OOC model [82]. OOCs are even compared across organs for the integration and evaluation of increasingly popular multi-OOC systems. BAP can provide assistance with research designs, including selecting or designing organ models, selecting compounds, exploring the properties of the compounds, and selecting cells for building models. Finally, on the add study page, the researchers set up the control group and the experiment group according to the study requirements and set each group’s type and quantity of chips, concentration, and addition time of compounds, cell samples that cultured, detection targets, and methods and other settings in detail. An example of the study design process is introduced in the literature [82]. In this example, the four-cell polyculture human liver MPS model SQL-SAL 1.0 was selected to test the hepatotoxicity of drugs. The detailed introduction page of the model can be accessed at https://mps.csb.pitt.edu/microdevices/model/2/ (accessed on 23 September 2022). The aim of this study was to compare the risk of hepatotoxicity of two similar drugs, Tolcapone and Entacapone. The design process first checked the compound details page of the two drugs and compared their molecular structure and chemical properties. Then, the adverse events (AE) data in the OpenFDA database was searched to obtain the frequency of adverse liver reactions of the two drugs in clinical use. Additionally, preclinical data from animal studies can also be accessed in the BAP. The inhibition rate of the two drugs was compared using a rat model. Finally, the authors configured a study in the BAP which includes 3 groups and 10 chips.
(ii)Data analysis.

After the researchers complete the experiment, the data files need to be uploaded to the configured study in specific formats. BAP supports uploading data files in three formats. The first format is the file format named MPS import-friendly columnar (MIF-c) especially developed for the database. The essence of this file is a table. Each column in the table is an attribute of the data, such as chip ID, detection method, time, value, etc., and each row is an experimental readout. Since the readout or result of the OOC experiment has only two forms, values or images/videos, the MIF-c format also has two variants: MIF-c-a file corresponding to value readouts and MIF-c-i file corresponding to image/video results. The MIF-c-i file only stores the detailed description of the image/video, and the actual assay images need to be uploaded to the BAP by the database administrator. A detailed description and reference template for this format can be found on the help page of the BAP. The second format is an omics data file, which is used for omics data analysis. The third format is assay plate reader data. The assay plate reader data integration tool is developed on the website, which allows users to set the detection board diagram, upload detection metadata, and calibrate and process metadata.

The built-in data visualization tool of the BAP website visualizes the data files uploaded by users on the website, which is the basic function of BAP data analysis. C3.js and D3.js are two data visualization JavaScript libraries that are used to present data in BAP [82]. There are many forms of diagrams. Appropriate diagrams will be selected according to the requirement of data, including histograms, line plots, bar graphs, dendrograms, etc. The lists of data sets which are instantly searchable and sortable are displayed by the JQuery library DataTables [93].
(iii)Reproducibility analysis.

The reproducibility of experimental results can be used as an important index of the performance of the MPS model. Reproducibility refers to the ability of the same measurement object to remain unchanged under changing conditions [94]. In each study, reproducibility analysis covers every measurement index of each group. The statistical method for calculating the index is based on the intra-class correlation coefficient (ICC) and coefficient of variation (CV). The coefficient of variation, like the variance, represents the degree of dispersion of the data, but the dimensional differences caused by different measurement methods in different experimental groups are eliminated by dividing by the mean. ICC is a reliability coefficient used to evaluate the consistency or independence of the same measurement result with widespread use in physiology statistics [95]. ICC has different forms after development, so it is necessary to select an appropriate model according to the data [96]. The ICC calculated in BAP is a two-way random effect model. According to the maximum value of CV in the group and the calculation result of ICC, reproducibility is divided into three grades: excellent, acceptable, and bad.

Another analytic tool is the statistical power analysis methodology. Users can choose to calculate the power *p* to estimate the probability of significant differences between two groups or to calculate the required sample size *n* [97]. There are four options as the parameters: Cohen’s effect size ‘*d*’ [97], Glass’ effect size ‘Δ’ [98], and Hedges’ effect size ‘*g*’ and ‘*g **’ [99].
(iv)Computational modeling.

OOCs serve as future tools for studying human disease models, predicting the clinical efficacy of drugs and their toxicological and pharmacokinetic parameters. Providing relevant computational models is also a major goal of BAP [100]. The computational modeling capabilities in BAP are in the early stages of development, and the PBPK analytic tool that is already online is only able to predict the intrinsic clearance of drugs based on liver modeling. As of October 2021, among the 141 studies publicly available in BAP, only four are for PBPK types, and the data can be used by the tools in the computational model. The predictive modeling tool is still in development. In a controlled experiment to test the computational PK model, the clearance of diclofenac was measured using the LAMPS model [19] and human hepatocyte suspension cultures, respectively [100]. Based on readings of the parent compound as well as major metabolite concentrations 4-(OH)-diclofenac in the effluent samples, PK parameters, including intrinsic clearance, were predicted and compared with clinical measurements. The results indicate that the predicted PK parameter values using the LAMPS model are closer to the true clinical range. After more experimental data of MPS models are uploaded to BAP, more tools, such as the intestine and kidney models that are equally important in the absorption and elimination of drugs, will be integrated into the computational model function.

#### 4.1.2. Public Data Included in BAP

BAP designs the data accessibility permission structure. Data contributors fully control the MPS data they provide. Within a data group, three types of users can view the data: data group administrator, editor, and viewer. The data administrator can also choose to share the data with other users by designating the collaborator groups and access groups or allowing it to be public to all users. As of May 2022, there are 152 open-access MPS models and 156 open-access studies in BAP. Figure 6 counts the percentage of different categories of open OOC models in BAP. The top three organs in the quantity of MPS models are the liver, kidney, and brain, respectively. In the study list, the liver model also accounts for the majority. Furthermore, for several different simulation modes of chips mentioned above, the detection indicators included in the database have obvious classification. Table 2 details the detection targets for some of the publicly available data in BAP.

The membranous mode chips that contain experimental data in BAP include the WAT model, kidney tubules model, and the blood–brain barrier model. The goal of blood–brain barrier model study is to test the barrier function. The main detection method of the model is the measured value of FITC-labeled glucan. The formation of a barrier in the chip is judged by the content of brain-side collected effluent. Lactate dehydrogenase and other common indicators were used to detect cell activity. WAT model mainly detects adiponectin, lipid droplets, lactate dehydrogenase, and other indicators to verify the growth. The tubules model mainly detects the metabolic response of vitamin D, ammonia content (pH change), and imaging of cell survival.

The polyculture models with experimental data included in BAP mainly include liver chips and bone chips. The drug toxicity test of liver chips mainly detected albumin, bile outflow, blood urea nitrogen, lactate dehydrogenase, and other indicators reflecting liver function and cell activity. The liver model also develops a PBPK model, which detects the drug probe concentration of coumarin, diclofenac, phenacetin, phenolphthalein, terfenadine, and testosterone. Bone chips detected the osteocalcin, osteopontin, and expression of alkaline phosphatase and luciferase to verify the osteogenesis and tested the relative recovery rate of some antitumor drugs.

The study of muscle bundle chips has the largest data set, but the detection indexes are relatively simple, which is the maximum elongation.

Mixed-form chips include brain chips and tumor chips, which mainly detect Parkinson’s disease protein and tumor growth imaging.

### 4.2. The Overview of Ocdb

Another OOC/microphysiological system database that has been reported is a project undertaken by the State Key Laboratory of Bioelectronics of Southeast University [17]. Ocdb is a web database platform for OOC resources, which can be accessed at http://www.organchip.cn/ (accessed on 23 September 2022). Ocdb can retrieve a great quantity of literature, patents, open access data of drugs, toxics, and gene expression. These data are downloaded from PubChem, NCBI, Drugbank, ChEMBL, BRD, OpenFDA, CTD, EMA, and other public databases and are integrated by Ocdb. Finally, Ocdb provides researchers with a comprehensive and practical OOC-related data retrieval website. In addition to the retrieving and downloading function, Ocdb also provides three main functions: mathematical modeling, three-dimensional modeling, and citation mapping [77]. The function of mathematical modeling provides a tool called tissue enrichment analysis which uses hypergeometric tests to calculate the enrichment of tissue-specific genes. Users are requested to input the gene list and the custom expression data sets. After analysis and modeling by the web server, results will be presented on the web page in the form of images and tables for users to view or download. This function is constantly expanding, and more tools will be integrated to support OOC data analysis and modeling in the future. A three-dimensional model is a navigation tool based on a 3D human anatomical model. Models including the brain, male body, and female body have been embedded. Users can find various human organs and corresponding OOC literature in the anatomical model. Furthermore, the citation map is a tool to assist researchers in grasping the research hotspots of OOCs. For better visualization, the citation map can demonstrate the cited number of authors’ papers, the citation relationship between authors, and the number of papers published in publications. Figure 7 shows the framework and construction of Ocdb.

### 4.3. Comparison of the Two Databases

BAP is a useful and well-established website for OOC users in North America [100]. Conversely, Ocdb provides services for Asia Pacific users and has features that BAP does not. Ocdb has a more powerful function in finding OOC-related literature and patents. Ocdb hosts 103,997 academic papers that can be searched by authors, institutions, and journals for different periods. Additionally, three-dimensional human anatomical models are used to subdivide the types of human organs related to OOCs. Compared with Ocdb, the references in BAP are presented as a subset of the MPS model or as a study set with a small number of papers and single sources, and search functions are lacking. Both databases have built-in mathematical modeling functions, but the tools they provide are aimed at different models. Ocdb provides a specific tissue enrichment gene test tool, whereas BAP is focused on PBPK modeling tools. Both Ocdb and BAP collect data from public databases that are helpful for OOC research, such as compounds and drugs, and have set up research direction hot spot queries. However, as an OOC database, Ocdb temporarily does not store OOC models or study readouts, nor does it compare models or analyze experimental data. Establishing a model database and collecting experimental data are the future directions for Ocdb development.

## 5. Application Prospect and Future Development Direction of OOC and Its Database

### 5.1. Personalized Medicine

Precision medicine refers to individuals receiving treatment tailored to their needs after modeling and stratification according to their own genetic, biosensor, clinical, and lifestyle data to promote, maintain, and restore their health [101]. OOCs have good application prospects in the field of precision medicine because OOCs are suitable for the personalized control of relevant components and parameters according to relevant personal health data [102]. For example, primary samples, such as patient blood, can be perfused on a vascular chip to monitor the formation of thrombosis to evaluate the potential drug response in individual patients [103]. Using patient iPSCs as the cell source, the OOCs created using certain patients’ cells more accurately represent the patient’s organ than any animals [102]. In particular, platforms similar to Biowire using patient-derived cell arrangements may be used to model rare genetic diseases [47]. The challenges affecting the application of personalized OOC mainly include the acquisition and modeling of personal samples and health data and the accuracy of prediction by these chips; a large amount of testing would be required to validate their usefulness [102]. An OOC database may help to promote the application of OOCs in precision medicine as the information of personalized models and silicon-calculated results could be generated and compared.

### 5.2. Pharmacology

Absorption, distribution, metabolism, and excretion (ADME) are key parameters in pharmacokinetics (PK). Using a quantitative method to predict the ADME parameters of controlled therapeutic drugs is a key challenge in drug development [104,105,106]. PK and pharmacology (PD) models are indispensable in the clinical trials of drugs. PK modeling has been developed to distribute and multicompartment physiological-based PK (PBPK) modeling and simulation methods to predict PK parameters using in vitro data. OOC models that can be applied to PBPK model analysis have been developed [107,108], and using OOC experimental data for in vitro to in vivo extrapolation (IVIVE) has been proposed. The data in Table 3 show that many studies in the field of PBPK started being published 10 years ago, and new works have been continuously published since then. The OOC field received little attention a decade ago, but the number of studies has substantially increased in the last five years. The combination of the two has only been witnessed in the literature in recent years, showing that to some extent, PBPK is a popular field that is developing, OOC is an emerging field, and the two have started to combine in the last five years. Importantly, BAP has already launched the computational modeling function for the MPS model for PBPK purposes, and subsequent prediction model is under development.

### 5.3. Environmental Toxicology

Environmental pollution directly affects human quality of life and production activities, threatens human health, and leads to a series of health problems. Compared with animal and 2D in vitro models, OOC models can more closely simulate the complex in vivo environment and real-time tissue interaction. Large throughput, clean testing, avoiding the use of animals, and fast response are the main advantages of OOC models over existing models [109,110]. The U.S. Environmental Protection Agency (EPA) stated that OOCs will be used to replace all mammal testing models by 2035 [111]. The OOC database will also provide unprecedented data analysis tools for toxicology usage in the near future.

### 5.4. Space Medicine

The mechanism of these disorders and the drugs for treatment are still being investigated. In 2016, CASIS and NIH/NCATS funded a space tissue chip program to enable scientists to better understand the role of microgravity on human health and diseases by designing tissue and organ chips and sending them to the international space station [112] as astronauts’ long-term exposure to microgravity and space radiation may lead to reversible effects on the human body, including muscle degeneration, osteoporosis (bone loss), orthostatic intolerance, the decline of cardiopulmonary function, and immune deficiency [113,114,115,116,117,118,119,120]. China and various European countries are also actively engaged in the R&D of aerospace-related OOCs. The developed space OOC system will be used to study the mechanism of changes in astronauts’ organ physiological functioning and the implementation of corresponding countermeasures. In this process, a large amount of data will be generated, and models will be created, so the OOC database can be used to support the data storage and analysis in these space medicine projects.

## 6. Conclusions and Future Perspectives

OOC is a cutting-edge technology in current biomedical research and pharmaceutical industries. For OOC models, their design, structure, and fabrication process, including devices, materials, flow path, pattern design, microsensors, cell sources, and measured data should all be properly recorded and analyzed. Additionally, their classification from the organ perspective and the simulation mode perspective should be well characterized. In addition, the relevant literature, as well as data on compounds and toxicants, should be stored. As a result, properly designed OOC databases, such as BAP and Ocdb, are essential for the future development of OOCs. BAP is currently working on the development of computational models for PBPK modeling on OOC, while Ocdb is developing the storage of OOC models and experimental data. Both databases currently provide limited support for image data, with only viewing capabilities, so better uploading functions are required by site managers. We think that, in the future, OOC databases will be able to integrate more manipulation methods for images, such as knowledge annotation to support the development of OOCs combined with AI or may even directly integrate AI models to provide features such as target recognition. These features may eventually lead to automated OOC databases to advance the field. In addition, the increasing complexity of OOC will pose challenges for OOC databases. With the emergence of new designs, current database modeling may not be able to store the OOC model completely. Modularity is a possible solution to this problem. Modular design can reduce the difficulty of designing and fabricating complex OOC models and improve the scalability of the models. The different component types are abstracted into entities in the database. Modular production requires a large number of tests, and more data are required to clarify the potential impact of modularization. The database will play an important role in collecting data and validating the whole modularization process.

## Figures and Tables

**Figure 1 bioengineering-09-00685-f001:**
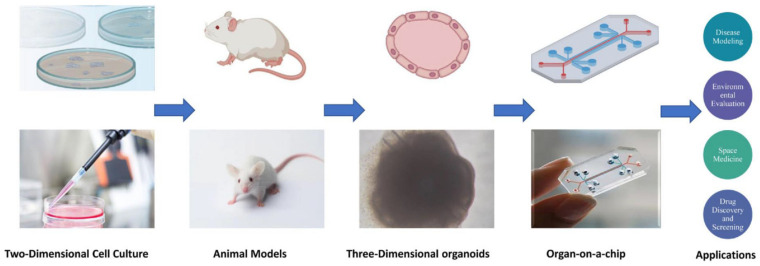
Evolution of the models and the application of OOC. From two-dimensional cell cultures to animal models and three-dimensional organoids to OOC, technology has evolved to allow more options for applications such as disease models and drug screening.

**Figure 2 bioengineering-09-00685-f002:**
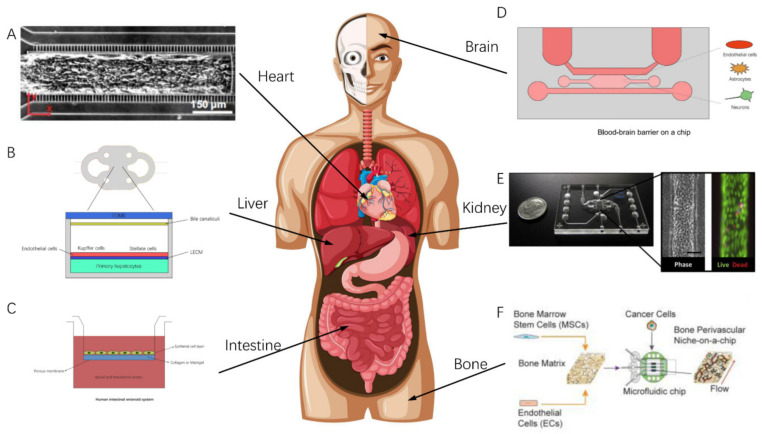
Some of the OOCs that have been developed for corresponding organs of the human body. (**A**) Cardiac MPS from UC Berkeley Healy Lab (adapted and modified from [25]). (**B**) The Liver Acinus MicroPhysiology System. (**C**) Intestine-on-a-chip. (**D**) Schematic of the human-neurovascular-unit-on-a-chip with a functional blood–brain barrier. (**E**) Kidney Proximal Tubule MPS (adapted and modified from [28]). (**F**) Schematic diagram of the preparation of the bone perivascular (BoPV) niche (adapted and modified from [38]).

**Figure 3 bioengineering-09-00685-f003:**
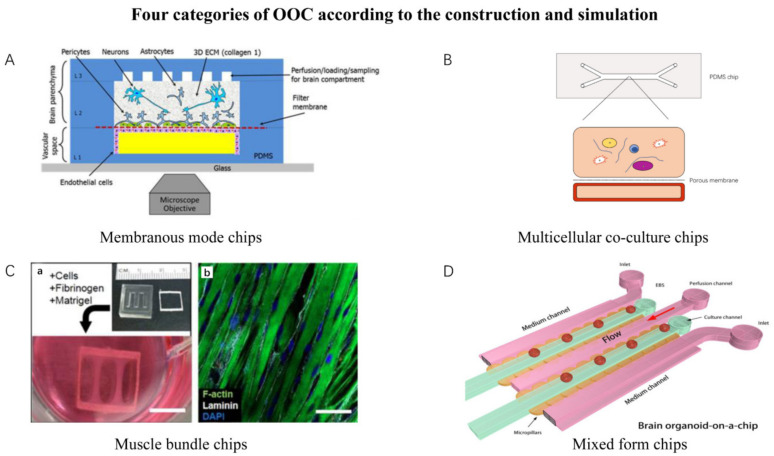
Four OOC models correspond to four types of simulation modes. (**A**) Blood–brain barrier chip, a representative of membrane mode chip (adapted and modified from [21]). (**B**) Schematic diagram of multicell culture in PDMS chip. (**C**) Human myogenic precursors were cultured in the nylon chip, a representative of a muscle bundle chip (adapted and modified from [30]). (**D**) Brain organoids-on-a-chip, a representative of mix form chip.

**Figure 4 bioengineering-09-00685-f004:**
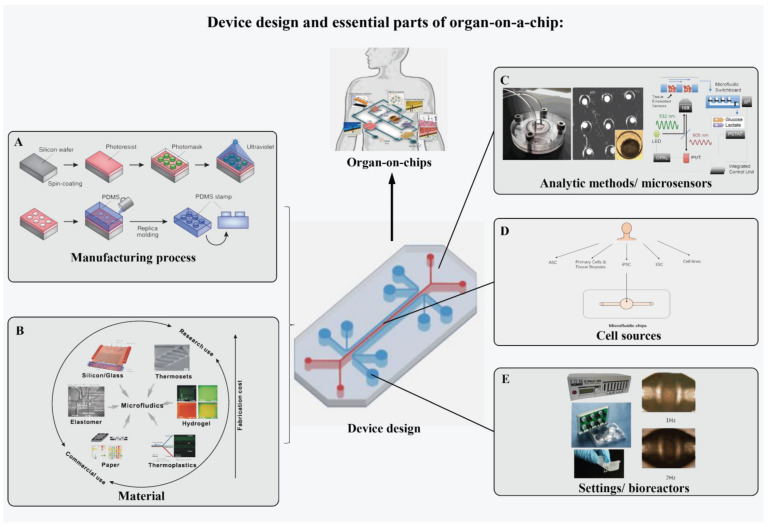
Schematic of device design and essential parts of OOC. (**A**) Manufacturing process of the device. A uniform layer of photosensitive material is spin-coated on the silicon wafer and then covered with a photomask with a microscale pattern. The microscale pattern etches into the photoresist after exposure to high-intensity ultraviolet light. PDMS stamps are obtained by casting the PDMS liquid prepolymer onto the etched photoresist pattern (adapted and modified from ref. [4]). (**B**) Materials of microfluidic chips (adapted and modified from ref. [49]). (**C**) A sensor system for metabolic monitoring. Nanoparticles embedded in the microtissues measure the Oxygen and pH. Aerometric biosensors downstream connected by a microfluidic switchboard measure the glucose and lactate (adapted and modified from ref. [50]). (**D**) Adult stem cells (ASCs), primary cells, cell lines, induced pluripotent stem cells (iPSCs), and embryonic stem cells (ESCs) can be differentiated and incorporated into microfluidic chips. (**E**) Cardiomyocytes are further differentiated and matured under the tension of microcantilevers and electrical stimulation, eventually achieving synchronized beating controlled by electrical stimulation.

**Figure 5 bioengineering-09-00685-f005:**
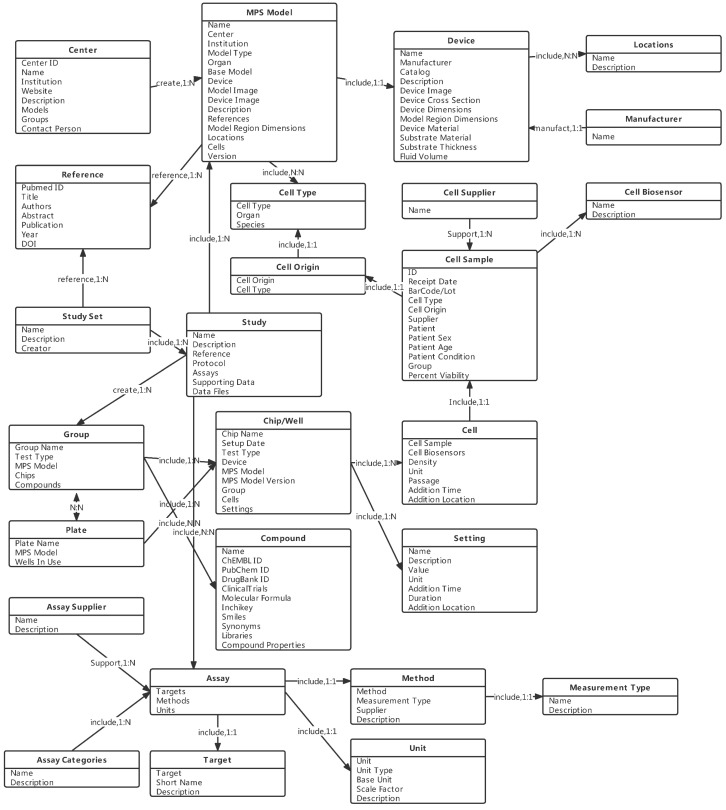
Entity relationship diagram abstracted from BAP website and OOC structure and classification. The rectangles in the figure represent entities, i.e., data objects in the data model. Each entity consists of its name and a set of attributes. The arrows between the entities represent the relationships between them, which are labeled with the type of relationship as well as the number. For example, devices are included in the MPS models, and each device entity forms dependencies with only one MPS model entity. The set of entities marked with different colors can be divided into different modules of the site.

**Figure 6 bioengineering-09-00685-f006:**
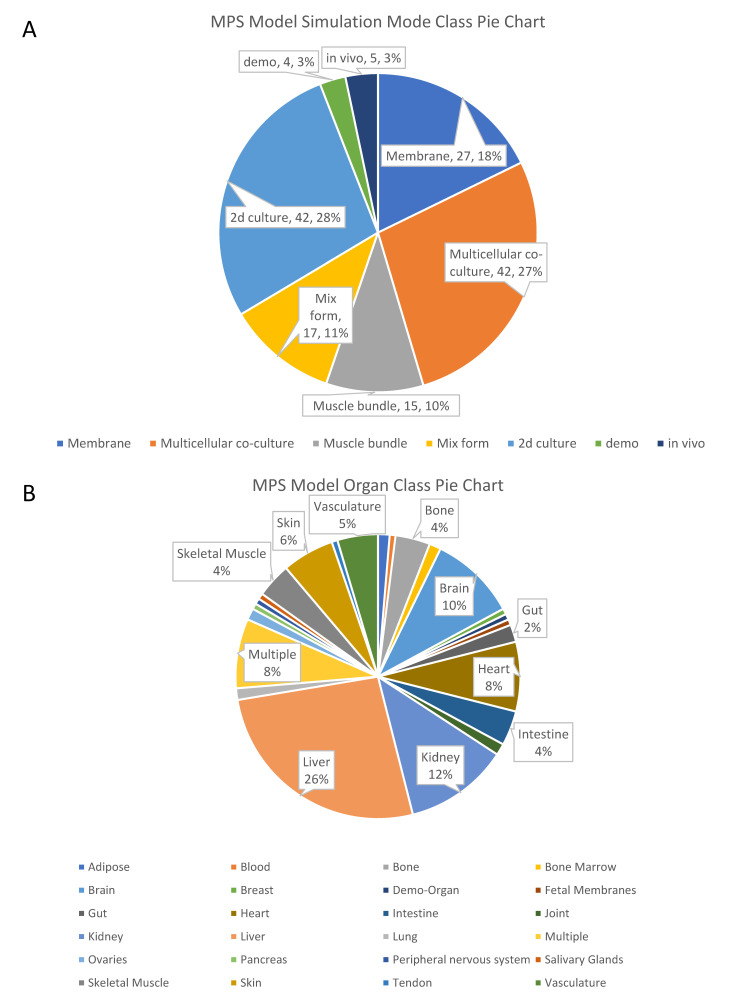
Pie charts of public MPS model data in BAP. (**A**) Pie chart of MPS model according to simulation mode classification. In order to be used as a control group in the study, the BAP has 28% of the total number of two-dimensional culture models. In addition, there is a small number of demos for demonstrating the function of the website. Among the four simulation modes, the multicellular co-culture mode accounts for the largest number. (**B**) Pie chart of MPS model according to organ classification. Classified by simulated organs, there are 24 types of public OOC models, of which the largest proportion is the liver-on-chips with 26%, and the second largest proportion is the kidney-on-chips with 12%. The percentage of organs that did not display data labels is 1%.

**Figure 7 bioengineering-09-00685-f007:**
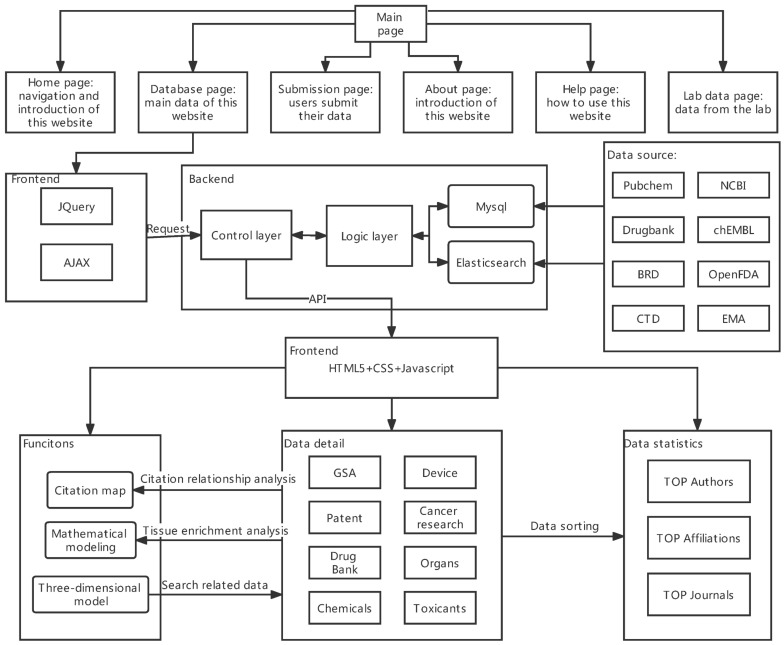
The main framework and construction of Ocdb. Six pages can be visited through the navigation bars, the most important of which is the database page. Based on the data the user wants to access, the front-end sends a request to the back-end. The back-end logic layer extracts and processes the data from MySQL and Elasticsearch, and the control layer distributes the requests and responses. The data sources of the Ocdb include Pubchem, NCBI, Drugbank, ChEMBL, BRD, OpenFDA, CTD, and EMA. Users can access three main functions, data details, and data statistics on the database page.

**Table 1 bioengineering-09-00685-t001:** Features and representatives of four categories.

Simulation Mode	Features	Representative Chips
Membranous mode chips	The chip has a layer of microporous membrane or semi-permeable membrane to connect two independent chambers.	Neurovascular unit on a chip. White adipose tissue on a chip.
Multicellular co-culture chips	The chip has one or multiple chambers for multicellular co-culture while allowing the cells’ spontaneous self-organization and interaction.	Liver Acinus Microphysiology System. Bone marrow on a chip.
Muscle bundle chips	Chips that culture muscle bundle.	Skeletal muscle culture system. Bioware II.
Mixed-form chips	Chips that involve multiple components together. Majority of organoids-on-chips could be classified into this category.	Brain organoid system on a chip. Tumor on a chip. Cardiac MPS.

**Table 2 bioengineering-09-00685-t002:** Detection targets of MPS models in the BAP.

Simulation Mode	MPS Model	Detection Target
Membrane Mode	Blood–brain barrier (NVU)	Dextran-FITC (10 kDa), Sigma-Aldrich: FD10S, Lactate Dehydrogenase, GM-CSF, IL-12/IL-23p40, IL-15, IL-16
	White Adipose Tissue	Bile Efflux, CYP3A4, Adiponectin, Lactate Dehydrogenase, Lipid Droplets, Lipid to Nuclei Ratio, Nuclei, PrestoBlue
	Intestinal Enteroid	FABP2/Human FABP2, Transepithelial Electrical Resistance (TEER), ATP, Dextran-FITC (10 kDa), Fexofenadine, Terfenadine
	Kidney Proximal Tubule	KIM-1 (human), Lactate Dehydrogenase (activity), 1α,25-Dihydroxyvitamin D, 24, 25-Dihydroxyvitamin D, 25-Hydroxyvitamin D, Gentamicin, Ammonium, Dead Cells, Human 1500+, PrestoBlue, Flowrate, Cadmium, Cisplatin
	Skin MPS	Corrosive, Non-Corrosive, Irritant, Non-Irritant, PrestoBlue, Lactate Dehydrogenase
Multicellular polyculture	LAMPs (SQL-SAL)	Albumin, Bile Efflux, Blood Urea Nitrogen, Lactate Dehydrogenase, COL Ia1, Fexofenadine, ROS, Terfenadine, TNF-Alpha, chenodeoxycholic acid, glycochenodeoxycholic acid, taurocholate, Steatosis, α-SMA, Coumarin, Diclofenac, Phenacetin, Phenolphthalein, Terfenadine, Testosterone, Caffeine, Pioglitazone, Rosiglitazone, Tolcapone, Troglitazone, Trovafloxacin, 4-hydroxydiclofenac, 6beta-Hydroxytestosterone, 7-Hydroxycoumarin glucuronide, Acetaminophen, Pioglitazone, E-Cadherin, anti-E-Cadherin, PrestoBlue, Flowrate
	Bone	Alkaline Phosphatase, Osteoprotegerin, Sclerostin, Osteocalcin, Osteopontin, Lactate Dehydrogenase, Luciferase Expression, Cisplatin, Dexamethasone, Methotrexate, Doxorubicin, Linsitinib
Muscle bundle	Skeletal Myobundle	Maximum Elongation
Mix former	Cardiac MPS	Beat Interval, Beat Rate, Contraction Velocity, Relaxation Velocity
	Brain	Park7-Recombinant (human), N-ACETYLASPARTIC ACID, Lactate Dehydrogenase
	Vascularized Tumor Model	Tumor Area, Tumor Growth, Tumor Integrated Intensity, Tumor Mean Intensity, Vessel Area, Vessel Junctions, Vessel Length, ATP

**Table 3 bioengineering-09-00685-t003:** Number of studies regarding OOC and PBPK in PubMed.

Search Keywords	Number of Studies in the Last 5 Years	Number of Studies in the Last 5–10 Years	Number of Studies from 10 Years Ago	Total
OOC	782	112	11	905
PBPK	3711	2559	4102	10,372
OOC AND PBPK	15	2	0	17

## Data Availability

Not applicable.

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
