# Peer review of "Organ-On-A-Chip Database Revealed—Achieving the Human Avatar in Silicon"

_bioengineering, 2022, doi:10.3390/bioengineering9110685_

Round 1

Reviewer 1 Report

This is a comprehensive review on OoC technology and databases that handle and manage data generated from OoC models. The authors also provide an extensive classification of OOC technology. I believe that the insight of the authors and the summary of relevant advances presented could be informative to researchers. The main issue that affects the quality of this work is the use of English language. 

Other comments:

- OoC technology also uses integrated sensors for monitoring of cell activity and function along with in line detection of analytes. The authors should also incorporate a section on that topic. Possibly before or after 2.2.4. 

- Explain the abbreviations.

- Other commetns:

Page 2 line 52: Place references since you mention "typical biomedical application".

Page 3 line104:  Rephrase the sentence. Correct "Linida". The name is Linda. Also it would be more appropriate to use "Griffith's group" or "Griffiths and coworkers...

Page 3 line142: remove the first name- Wang et al., 

Page 3 line145-148: Rephrase the sentence. Correct name is Linda not "Linida". Also it would be more appropriate to use "Griffith's group" or "Griffiths and coworkers...

Page 4 line 171: Rephrase the sentence. Correct "Linida". The name is Linda. Also it would be more appropriate to use "Griffith's group" or "Griffiths and coworkers...

Page 5 line 250: You should write the complete name i.e., colorectal biomarker database (CBD) 

Page 7 line 321: Use capital letters for the first letter. i.e., Medical Information Mart for Intensive Care (MIMIC). 

Author Response

Point 1: The main issue that affects the quality of this work is the use of English language.

Response 1: We have used the language editing service provided by MDPI. The article has been extensively revised in English.

Point 2: OoC technology also uses integrated sensors for monitoring of cell activity and function along with in line detection of analytes. The authors should also incorporate a section on that topic. Possibly before or after 2.2.4.

Response 2: We change the subheading of 2.2.4 to Sensory systems. With advances in the development of OOC platforms, there have been efforts to integrate sensors into OOC devices to monitor the performance of the tissue as well as the extracellular environment. These sensors can also detect target analytes. So we think that it is most reasonable to divide it into sensory system from the structure.

Point 3: Explain the abbreviations and other comments.

Response 3: More correct use of abbreviations in the article. Comments are met.

Reviewer 2 Report

The manuscript provides an overview of two databases covering in vitro models, including organ on a chip models, for pharmacological, as well as mechanistic studies. The standardization and collection different studies in comprehensive databases is of utmost importance for the progress and sound expansion of the field. The manuscript is therefore very timely and should be published after addressing the comments below:

·       Section 2 in its current state is somewhat limited. Maybe a broader introduction including what are the minimum criteria to consider an assay as organ on a chip could be stated. There are several microfluidic studies mimicking physiological or pathophysiologic conditions (shear rates, adhesion and chemokine molecules, etc.) without any on chip culture for monitoring patient status or probing blood cells.

·       Labeling of ‘membranous mode chips’ and ‘multicellular co-culture chips’ can be confusing since membrane-based models can also accommodate multicellular co-cultures and membrane material and structure can be tuned to match gel layers presented in the multicellular co-culture chips. One of the most visible distinctions is the vertical or in plane organization of compartments.

·       Numbers provided in the section 4. iii) should be updated.

·       Subtitles in section 4 should be restructured to reflect different databases more clearly.

·       Organ on chip studies are advancing rapidly with the addition of niche approaches, which could bring additional challenges in terms of standardization of the fabrication and measurement techniques. What are the authors’ opinions on how to deal with increasing complexity in such databases and further in silico modeling?

·       There are minor language errors and typos throughout the manuscript.

Author Response

Point 1: Section 2 in its current state is somewhat limited. Maybe a broader introduction including what are the minimum criteria to consider an assay as organ on a chip could be stated. There are several microfluidic studies mimicking physiological or pathophysiologic conditions (shear rates, adhesion and chemokine molecules, etc.) without any on chip culture for monitoring patient status or probing blood cells.

Response 1: It is difficult to define a minimum standard for organ-on-chips now. But the relevant standards will be gradually established. we explain the future of the chip will be standardized production in the article. While standardizing the manufacturing of basic chips, the application of chips to personalized medicine will require the addition of different modules to achieve different purposes. We classified and structurally dissected the existing organ-on-chips. This is in line with the industrialization of organ-on-a-chip and provides ideas for database storage.

Point 2: Labeling of ‘membranous mode chips’ and ‘multicellular co-culture chips’ can be confusing since membrane-based models can also accommodate multicellular co-cultures and membrane material and structure can be tuned to match gel layers presented in the multicellular co-culture chips. One of the most visible distinctions is the vertical or in plane organization of compartments.

Response 2: The classification is based on methodologies for OOCs construction and simulation. Due to the technology limitations, different OOCs usually share common modes. However, the number of OOCs that combine different simulation modes will increase in the future. Embodied in the database, an OOC may have different labels for filtering.

Point 3: Numbers provided in the section 4. iii) should be updated. Subtitles in section 4 should be restructured to reflect different databases more clearly.

Response 3: We adjusted the subheadings in section 4 to make it more intuitive.

Point 4: Organ on chip studies are advancing rapidly with the addition of niche approaches, which could bring additional challenges in terms of standardization of the fabrication and measurement techniques. What are the authors’ opinions on how to deal with increasing complexity in such databases and further in silico modeling?

Response 4: Modular design can reduce the difficulty of designing and fabricating complex OOC models, and improve the scalability of the models. Ronaldson-Bouchard, K., et al designed a multi-organ chip in which matured tissue niches are linked by recirculating vascular flow. They use a modular design, with four culture chambers that can each contain 1.5 ml of tissue-specific medium, and a reservoir for the recirculating vascular medium. The different component types are abstracted into entities in the database. The emergence of new structures may allow the addition of new tables to the database. But in most cases, the increasing complexity is caused by increasing number of similar components.

Point 5: There are minor language errors and typos throughout the manuscript.

Response 5: We have used the language editing service provided by MDPI. The article has been extensively revised in English.

Round 2

Reviewer 2 Report

The authors did not comprehensively address most of my original comments and provided handwavy explanations at most. Changes made in the manuscript should be provided along with the responses.

Point 1. The authors should address this issue in the manuscript. Are simpler microfluidic devices included in such databases? What is the authors’ critical opinion on how to integrate such “assays”?

Point 2. The authors should better clarify in each classification section what they mean and, since the focus of this manuscript is databases, they should critically discuss the potential problematic areas and solution over such classification issues in the manuscript.

Point 4. The authors should integrate their response into the manuscript as well. Also, even though the modularization approach sounds reasonable, inter-modular and systemic effects on different modules requires caution when comparing with individual OOCs.  

Author Response

Point 1. The authors should address this issue in the manuscript. Are simpler microfluidic devices included in such databases? What is the authors’ critical opinion on how to integrate such “assays”?

Response 1: We thank the reviewer for this comment. In fact, the BAP database included from static 2-D culture models to simple microfluidic models together with complicated 3D models. But simple models could not be classified as OOC model. To better explain this, we add an extra paragraph below section 2 to discuss this issue. We also cited the FDA's definition of OOC which describes the minimum standard for OOC. Thus, previous assays that do not meet the OOC standards have been classified as traditional models.

Point 2. The authors should better clarify in each classification section what they mean and, since the focus of this manuscript is databases, they should critically discuss the potential problematic areas and solution over such classification issues in the manuscript.

Response 2: We thank the reviewer for this suggestion. We added a table to show the four categories that we listed to make them look clearer. We also discussed the limitations of this classification in subsection 2.1. We fully understood that OOCs have multiple patterns, morphologies or functions, thus they may fit to different categories at the same time. Despite this problem, performing model classification at current stage – maybe with multiple labels –could in general benefit to models classification and retrieval.

Point 4. The authors should integrate their response into the manuscript as well. Also, even though the modularization approach sounds reasonable, inter-modular and systemic effects on different modules requires caution when comparing with individual OOCs.  

Response 4: We thank the reviewer for the suggestion. We have placed this discussion in section 6. The increased complexity of the OOC model poses a challenge to the database. Modularity is a possible solution to this problem. It is worth noting that modular production also requires a large number of tests, and more data are required to clarify the potential impact of modularization. The database will play an important role in collecting data and prove/validate the whole modularization process.